# Env from EIAV vaccine delicately regulates NLRP3 activation via attenuating NLRP3-NEK7 interaction

Xing Guo[1☯], Cong Liu[1☯], Yuhong Wang[2], Hongxin Li[1], Saiwen Ma[1], Lei Na[1], Huiling Ren[1], Yuezhi Lin[1]*, Xiaojun Wang [1,3]*

**1** State Key Laboratory for Animal Disease Control and Prevention, Harbin Veterinary Research Institute of Chinese Academy of Agricultural Sciences, Harbin, China, **2** Department of Geriatrics, First Affiliated Hospital of Harbin Medical University, Harbin, China, **3** Institute of Western Agriculture, Chinese Academy of Agricultural Sciences, Changji, China

☯ These authors contributed equally to this work.
* Linyuezhi@caas.cn (YL); wangxiaojun@caas.cn (XW)

## Abstract

The current equine infectious anemia virus (EIAV) vaccine causes attenuation of the inflammatory response to an appropriate level, compared to that produced by virulent EIAV. However, how the EIAV vaccine finely regulates the inflammatory response remains unclear. Using a constructed NLRP3-IL-1β screening system, viral proteins from two EIAV strains (the attenuated vaccine and its virulent mother strain) were examined separately. Firstly, EIAV-Env was screened to direct binding P2X7 (R) with notable $K^+$ efflux trans-cellularly. Secondly, EIAV-Env was found to bind NLRP3 and/or NEK7 to trigger aggregation of NLRP3-NEK7 to form NLRP3-NEK7 complex in cells. Comparison of the two strains, we observed a significant reduction on vaccine-Env-initiated NLRP3-NEK7 complex formation, with no difference in Env triggering P2X7 (R)-mediated ion fluxes. Thirdly, reciprocally mutation on four stable varied amino acids between two strains produced an anticipated outcome on NLRP3-IL-1β-axis activation. As the attenuated vaccine was shown evolved as a natural quasispecies of the virulent EIAV, its precise and adaptable regulation via spatial proximity-dependent intracellular activation might present a "win-win" virus-host adaption, offering an alternative strategy on envelop-based vaccines development.

## Author summary

Here, we report that EIAV-Env mediates NLRP3 inflammasome activation through two distinct pathways. The first pathway involves a transcellular mechanism driven by $K^+$ flux, which couples Env-P2X7 interaction. The second pathway entails direct intracellular binding between Env and NLRP3, promoting the assembly of NLRP3-NEK7 and subsequent inflammasome formation. Notably, we

**Data availability statement:** The authors confirm that all data underlying the findings are fully available without restriction. All relevant data are within the paper and its Supporting Information files.

**Funding:** This study was supported by grants from Heilongjiang Provincial Natural Science Foundation of China (LH2023C049 to YL), the Natural Science Foundation of Heilongjiang Province (TD2022C006 to XW), the National Natural Science Foundation of China (32372985 to YL), and the National Key Research and Development Program of China (2023YFD1802500 to YL). The funders had no role in study design, data collection and analysis, decision to publish, or preparation of the manuscript.

**Competing interests:** The authors have declared that no competing interests exist.

observed a marked difference in NLRP3 inflammasome activation between the vaccine and virulent strains, which was reflected in the extent of Env-mediated NLRP3-NEK7 aggregation. This study not only enhances our understanding of lentivirus-host immune interactions but also contributes to the broader discourse on virus evolution and host-induced inflammation.

## Introduction

Equine infectious anemia virus (EIAV) belongs to the lentivirus family, and shares similar virus-host immunity and pathogenesis with the human immunodeficiency virus (HIV-1) [1–4]. Despite extensive efforts, safe and sustainable HIV-1 vaccines have yet to be developed, primarily due to the presence of latent virus reservoirs and rapid viral evolution [5–7]. In contrast, an attenuated vaccine against EIAV, derived from its virulent progenitor strain, has been successfully developed and widely utilized for decades [3,8]. Notably, natural EIAV quasispecies in equine populations have also been observed to undergo attenuation with time [9–12]. Through systemic sequencing, our "artificially" attenuated EIAV vaccine has been demonstrated to share similar evolutionary traits with the naturally attenuated EIAV [8,13]. This suggests that the vaccine can be regarded as a "natural" EIAV that has undergone accelerated evolution.

Several studies have demonstrated that the virulent EIAV strain elicits more pronounced inflammatory responses and pathological effects than those produced by the attenuated EIAV vaccine [4,14]. Specifically, there are significant differences observed in the production levels of interleukin-1 beta (IL-1β), a cytokine tightly regulated by the inflammasome, between the two EIAV strains. The NLRP3-IL-1β inflammatory pathway is well-documented in viral pathogenesis, often triggered by various pathogens through P2X7-dependent signaling [15]. A vast number pathogens are able to trigger NLRP3-IL-1β-axis activation via P2X7-dependent signaling [16–19]. However, it's noteworthy that certain pathogens activate the NLRP3 inflammasome through ATP- and P2X7-independent pathways [20,21]. Despite these insights, the mechanism by which EIAV activates the inflammasome and subsequent inflammatory pathways remains unclear. Therefore, the aim of the study is to delve into the molecular profiles of inflammasome activation induced by both the attenuated vaccine strain and the virulent mother strain of EIAV. This investigation seeks to uncover the reasons behind the differential regulation of the inflammasome by these two EIAV strains. Our findings may provide a potential target for fine-tuning the regulation of the NLRP3-IL-1β axis, which could have implications for the development of safe and efficient envelop-based vaccines.

## Results

### The attenuated vaccine triggers markedly lower NLRP3-IL-1β-axis activity than the virulent EIAV

The utilization of the semi-quantitative scoring platform, revealed a notable reduction in inflammatory lesions within the vaccine group compared to the virulent group

(S1 Fig and Table 1) [22]. This finding highlights the potential effectiveness of the vaccine in attenuating inflammatory reactions and pathological harm associated with EIAV infection. Moreover, previous studies have demonstrated significant differences in the presence of IL-1β following infection with the EIAV vaccine compared to the virulent strain [4,23]. These findings further strengthened the understanding of the vaccine's impact on inflammatory responses and disease progression. Therefore, we further compared the molecular regulation of IL-1β associated pathways by the EIAV vaccine strain and the EIAV virulent strain parallelly *in vivo* and *in vitro*. *In vivo* infections resulted in a consistent elevation of IL-1β induced by EIAV vaccine or by virulent strain, with much lower trajectory induced by vaccine (Fig 1A). *In vitro* studies in equine macrophages showed a similar difference between the attenuated and virulent EIAV strains (Fig 1B). Interestingly, the transcription level of pro-IL-1β did not significantly differ between the two groups, indicating that the vaccine robustly regulated the conversion of pro-IL-1β to mature IL-1β (Fig 1C). The NLR family proteins play a key role in the regulation of the conversion of pro-IL-1β into mature IL-1β, and we therefore compared the expression levels of NLRP1, NLRP3 or NLRC4 mRNA between the two groups using quantitative RT-PCR. We found that NLRP3 mRNA was significantly enhanced in macrophages following infection with either virulent EIAV or the attenuated vaccine, with no significant differences between the two strains. Moreover, the expression of NLRP1 and NLRP4 mRNA was not induced following infection with either strain of EIAV (Fig 1D). We also visualized the formation of ASC speck, a hallmark of inflammasome activation, in macrophages infected with either the attenuated vaccine or with virulent EIAV (Fig 1E). Using inhibitor (MCC950) to inhibit NLRP3-IL-1β signaling, the expression of IL-1β induced by EIAV was remarkably down-regulated with a similar level as negative control (Fig 1F), further confirming that the NLRP3-IL-1β-axis had been activated by both the virulent strain and by the attenuated vaccine.

## Screening for EIAV-Env component able to activate the NLRP3-IL-1β-axis interactively

We observed that the activation of the NLPR3 inflammasome and the levels of IL-1β appeared significantly difference between the two EIAV strains. We therefore wanted to screen for any component of EIAV that interacted with any molecule within the NLRP3-IL-1β-axis. We constructed an equine NLRP3-IL-1β-axis screening system (Fig 2A), and found that the viral Env protein was the dominant component of EIAV that induced IL-1β secretion with a dosage-dependent effect in this system (Fig 2B). Furthermore, we confirmed the subunit of Env (Gp90) that activated the NLRP3 inflammasome (Fig 2C). We then visualized the ASC speck formation in *env*- or *gp90*-transfected 293T cells with a confocal microscope to further validate these findings (Fig 2D). Then equine macrophages were infected with two Env-pseudotyped viruses. We observed that secretion of mature IL-1β was induced by both of the Env-pseudotyped viruses, confirming that Env could

**Table 1. Representative images of inflammatory pathological lesions in lung, liver, spleen, kidney and lymph node (*20) from three equines inoculated with EIAV attenuated vaccine or from three equines infected with EIAV virulent strain.**

| Group | Animal ID | Scoring on inflammatory pathological lesions | | | | |
|---|---|---|---|---|---|---|
| | | Lung | Liver | Spleen | Kidney | Lynph node |
| Vaccine strain | Vac-1 | 0 | 0.5 | 0 | 0 | 0 |
| | Vac-2 | 0 | 1 | 0 | 0 | 1 |
| | Vac-3 | 0.5 | 0 | 0 | 0.5 | 1 |
| Virulent strain | Vir-1 | 3 | 2.5 | 2.5 | 2 | 2.5 |
| | Vir-2 | 2.5 | 2 | 2 | 2 | 3 |
| | Vir-3 | 3 | 1.5 | 3 | 1 | 2 |

Note: Using the semi-quantitative scoring method that we had applied, inflammatory pathological lesions in five organs from three equines that inoculated with attenuated vaccine or from three equines that infected with EIAV virulent strain were scored. "0" indicated a normal organ,"1" indicated mild pathological changes, "2" indicated moderate pathological changes and "3" indicated severe pathological changes. 0.5 score was applied for more delicate comparison in the present study with agreement of two professional pathologists.

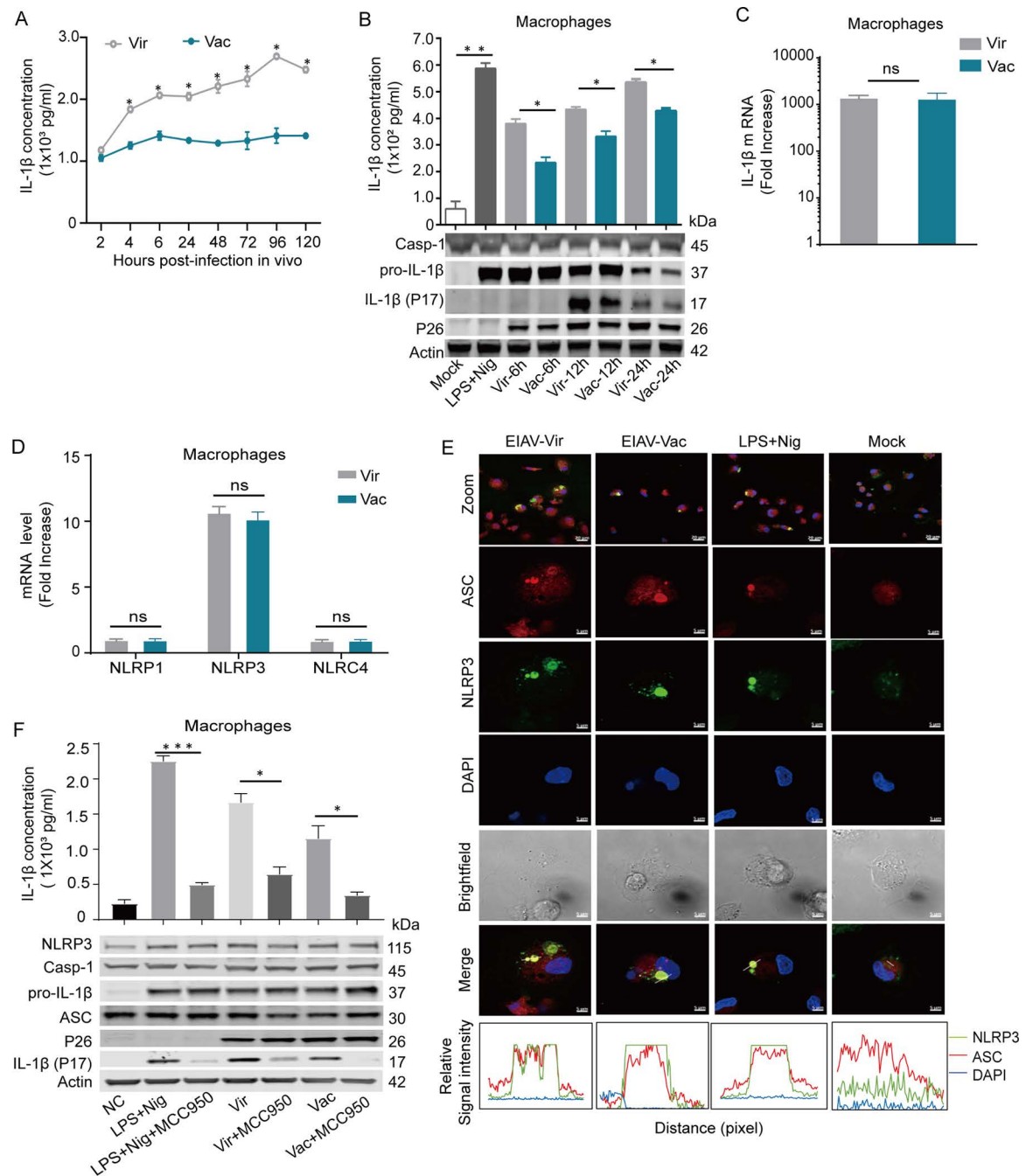

**Fig 1. The vaccine EIAV induces markedly lower levels of IL-1β than the virulent EIAV *in vivo* and *in vitro*. (A)** Quantification of IL-1β levels in the sera of individuals infected with virulent EIAV and vaccine EIAV using ELISA assay (n = 2 per group). **(B)** Quantification of IL-1β in the supernatants of virulent and vaccine EIAV-infected macrophages. Western blot analysis addressing expression of pro-IL-1β and mIL-1β was performed parallelly. Results are mean of 2 independent experiments performed with eMDMs of 2 donors. **(C-D)** Quantification of mRNA expression of IL-1β (C) and NLR family genes (NLRP1/NLRP3/NLRC4) (D) in macrophage cells infected with virulent or vaccine EIAV strains using real-time RT-PCR. **(E)** Visualization of endogenous NLRP3 (green)-ASC (red) puncta formation in macrophages infected with virulent or vaccine strain (12 h *p.i.*) using confocal microscopy (Nigericin (Nig)-treated macrophages served as the positive control). **(F)** Representative images and statistical analyses from Western blot and ELISA demonstrate the expression levels of IL-1β in equine macrophages treated with MCC950, along with virulent or vaccine EIAV strains (2 x $10^5$ TCID$_{50}$, 24 h), or with LPS (10 mg/mL, 6 h) and Nigericin (10 µM, 2 h) as control. In C-F, results are mean of 3 independent experiments performed with eMDMs of different donors (n = 3 per group).

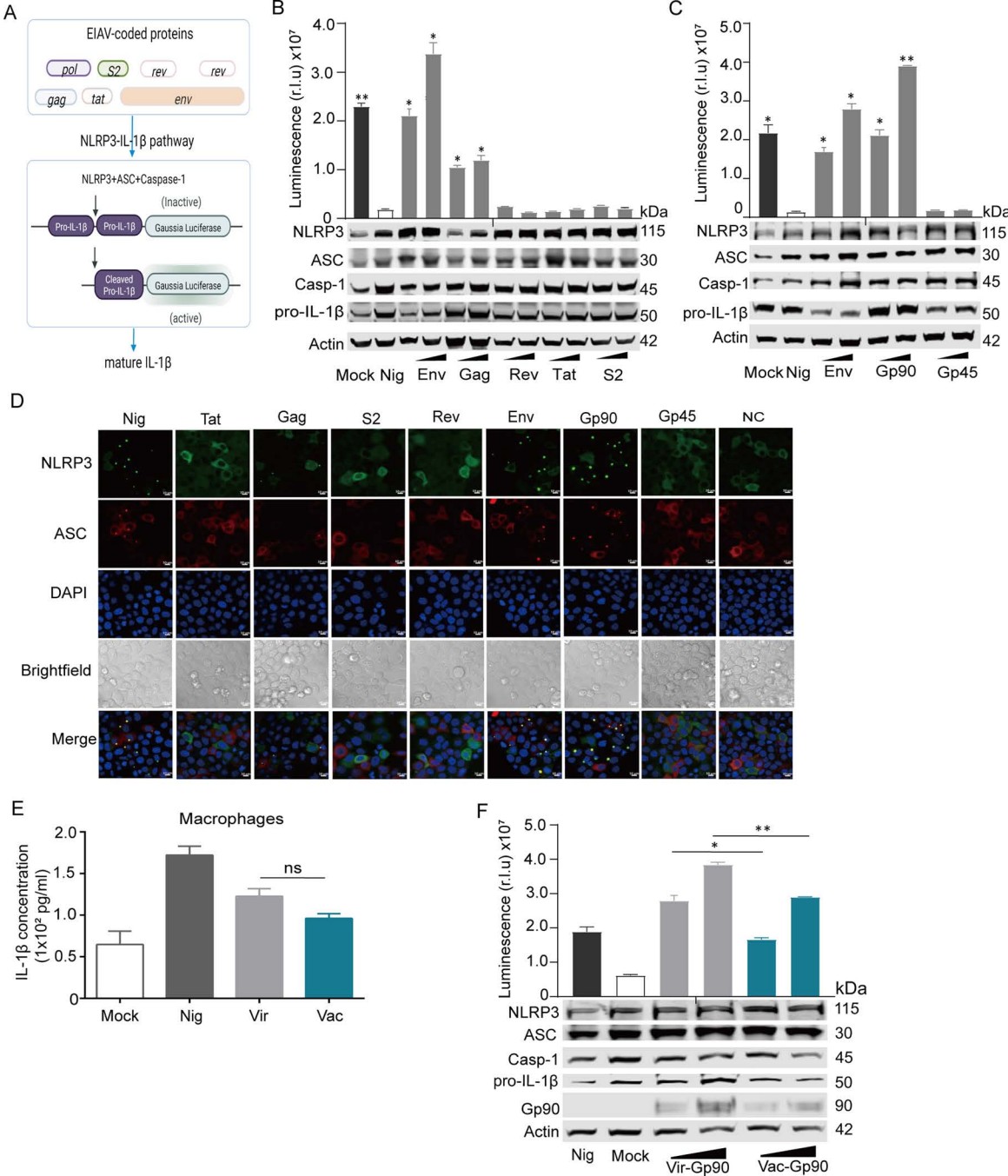

**Fig 2. Screening for EIAV-coded proteins involved in the NLRP3-IL-1β-axis activation. (A)** Schematic of the bioactivity of the NLRP3-IL-1β signaling in 293T cells co-transfected with EIAV-coded proteins *via* a luciferase reporter system. **(B)** 293T cells were co-transfected with the indicated reporter constructs in the presence or absence of EIAV expression plasmids separately. Luciferase activity and reporter construct expression were analyzed as in **(A)**. **(C)** Cells were subjected to the same assay protocol as **(B)** but were transfected with *gp90* and *gp45* separately. Analysis was performed as described in **(A)**. **(D)** Con-focal microscopy visualization of the formation of NLRP3-ASC puncta (yellow, indicated by white arrows) in 293T cells co-transfected with one of the five EIAV-coded proteins (Env/Gag/S2/Rev/Tat) or with one of the two Env glycoprotein components (Gp90 and Gp45). **(E)** Statistical analysis of ELISA data was conducted to evaluate the expression levels of IL-1β in macrophages infected separately with Env-pseudotyped viruses derived from virulent or vaccine EIAV strains at 24 hours post-infection. **(F)** Same as (B) but co-transfected with virulent-*gp90* and vaccine-*gp90* separately (* $P < 0.05$, ** $P < 0.01$, compared to the mock-transfected group). All data are mean of 2 independent experiments (n = 2 per group).

mediate NLRP3-IL-1β-axis activation (Fig 2E). Next, we sought to determine if vaccine-Env and virulent-Env differed in NLRP3-IL-1β-axis activation. 293T cells were co-transfected with Env (s) and plasmids of NLRP3-IL-1β screening system. We observed significantly lower IL-1β production in 293T cells transfected with vaccine-*gp90*, compared with that in 293T cells transfected with virulent-*gp90* (Fig 2F). From these analyses, we speculated that Env was the main factor affecting the effects of the NLRP3-IL-1β-axis activation induced by EIAV infection.

## Screening target cellular proteins involved in the activation of NLRP3-IL-1β-axis by EIAV-Env

To screen for target cellular protein (s) that are associated with NLRP3 inflammation activation by EIAV-Env, we performed an Env IP-mass spectrometry (IP-MS) (Fig 3A). Three target inflammasome-associated proteins, P2X7 (R), NEK7 and NLRP3, were shown to directly interact with Env in virulent EIAV-infected macrophages (Fig 3B and 3C). We further validated this result using a co-immunoprecipitation (Co-IP) assay. We observed that Env co-precipitated with P2X7 (R), NEK7 or NLRP3 in both vaccine-*env* or virulent-*env* transfected 293T cells (Fig 3D–3F). Co-localization of Env and P2X7 (R), NEK7 or NLRP3 in both groups could be visualized using confocal microscopy (Fig 3G–3I). Therefore, we speculated that the activation of the NLRP3 inflammasome induced by EIAV infection occurred *via* a mechanism dependent on Env/P2X7 (R)/NLRP3/NEK7 binding. Interestingly, no disparity between the abilities of Env to bind these proteins was observed between the virulent and vaccine EIAV strains.

## Investigation of the P2X7 (R)-regulated $K^+$/$Ca^{2+}$ channel status following infection with attenuated vaccine or virulent EIAV

The role of the Env-P2X7 (R) interaction in the regulation of the NLRP3 inflammasome was then explored. As an ATP-gated ionic channel [24,25], the opening of P2X7 (R) leads to low intracellular $K^+$ concentrations or high intracellular $Ca^{2+}$ concentrations, which then trigger the activation of the NLRP3 inflammasome [17,24]. We therefore examined whether or not the P2X7 (R) ionic channel was activated in EIAV-infected macrophages through comparison of intracellular $K^+$/$Ca^{2+}$ concentrations with or without EIAV infection. As excepted, we observed that intracellular $K^+$ concentrations rapidly decreased (Fig 4A) in macrophages infected by either the virulent or the vaccine EIAV strain, while the $Ca^{2+}$ concentrations remain stable or slightly increase in time (Fig 4B). Consistent with this result, P2X7 (R) knockdown prevented $K^+$ efflux in macrophages infected with attenuated vaccine or the virulent EIAV (Fig 4C). To further validate this in 293T cells induced by Env, we pretreated 293T cells using Glybenclamide (a $K^+$ inhibitor), or grew the cells on medium containing either high $K^+$ concentrations and then measured the activation of the NLRP3-IL-1β-axis. We observed that Glybenclamide (a $K^+$ inhibitor) significantly reduced NLRP3-IL-1β-axis activation in both attenuated Env and virulent Env (Fig 4D), a finding further supported by treatment with high $K^+$ medium (Fig 4E). These observations suggest that both virulent-Env and vaccine-Env are able to induce $K^+$ efflux-dependent NLRP3-IL-1β-axis activation. Notably, no differences were observed in the Env-mediated $K^+$ efflux between the two EIAV strains.

## Vaccine-Env attenuated the NLRP3-NEK7 complex compared with virulent-Env

To explore whether P2X7 (R) contributes to Env-induced NLRP3-IL-1β-axis activation, we generated 293T cell lines stably expressing P2X7 (R). The use of P2X7 (R)-specific siRNA and a P2X7 (R) inhibitor revealed a significant reduction in IL-1β production in cells treated with either the virulent-Env or vaccine-Env, compared to untreated groups. Interestingly, the observed reduction in IL-1β production was similar between cells treated with the attenuated vaccine-Env and those treated with the virulent-Env (Fig 5A and 5B), indicating that the involvement of P2X7 (R) in IL-1β regulation is consistent regardless of the virulence of the strain. By elucidating the ability of Env to directly bind NLRP3 or NEK7 in cells, our findings bolster the hypothesis of a P2X7 (R)-independent activation mechanism contributing to NLRP3 inflammasome activation during EIAV infection (Fig 3E and 3F). Following the co-transfection of three plasmids (*env*, *NLPR3*, and *NEK7*)

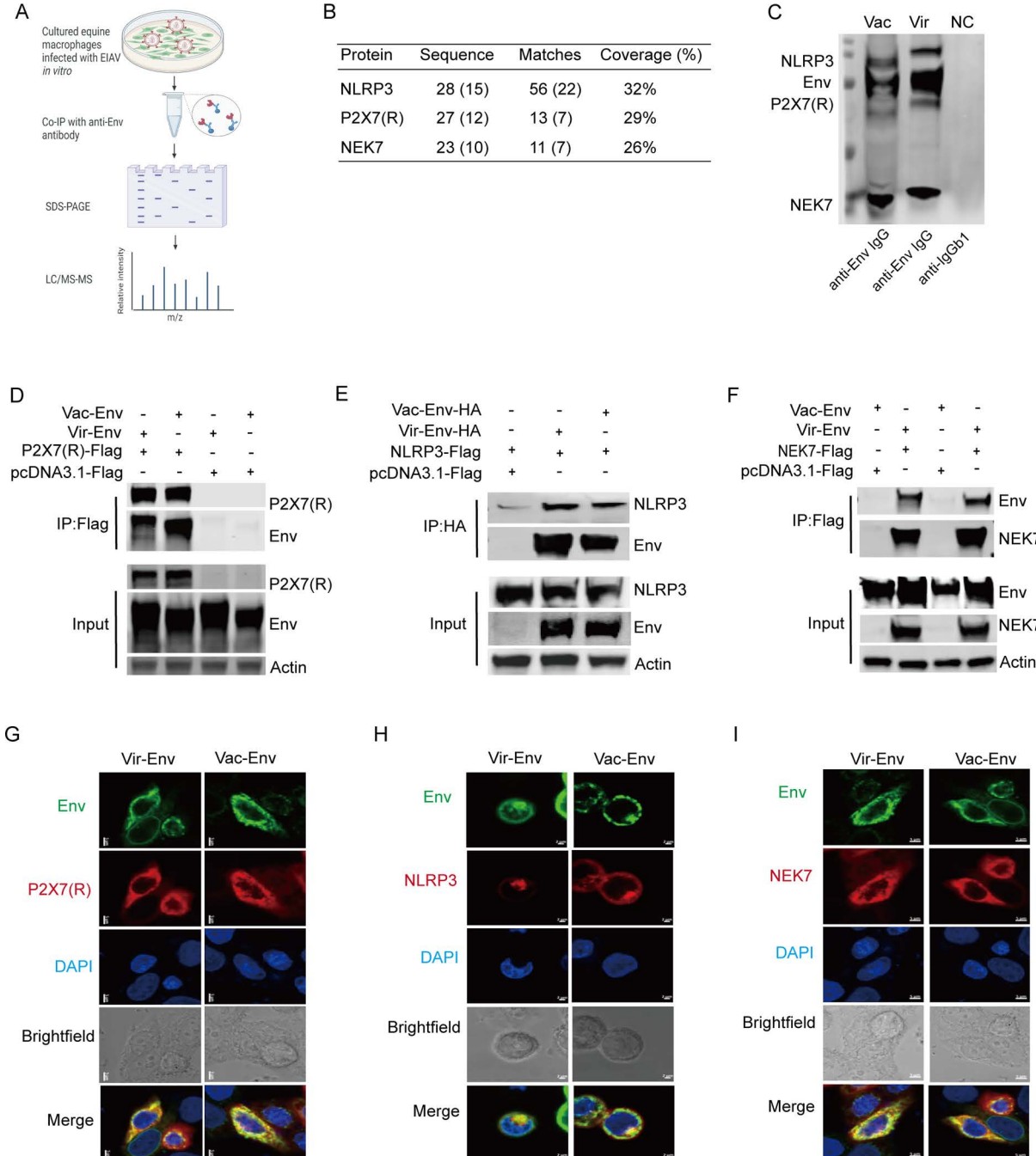

**Fig 3. Screening for Env-interacted cellular protein (s) involved in the NLRP3-IL-1β-axis. (A)** Schematic illustration of the IP-mass spectrometry screening for protein (s) in the NLRP3-IL-1β-axis that interact with EIAV-Env. Clip art created in BioRender Yuezhi L (2025). https://BioRender.com/i04f345. **(B)** Mass spectrometry analysis of NLRP3, P2X7 (R), and NEK7 peptides following EIAV infection. **(C)** Immunoprecipitation with anti-Env antibodies of endogenous P2X7 (R), NLRP3 or NEK7 in macrophages infected with EIAV stain. **(D)** Co-immunoprecipitation of HA-Env with Flag-NLPR3 in 293T cells transfected with the indicated expression plasmids. **(E)** Same procedure as (D) but cells transfected with Flag-NLRP3 and Flag-*env* (s). **(F)** Same procedure as (D) but cells transfected with Flag-NEK7 and Flag-*env* (s). **(G-I)** Co-localization between Flag-P2X7 (R) and HA-Env, Flag-NLRP3 and HA-Env or Flag-NEK7 and HA-Env was examined using confocal microscopy. Scale bars: 10 μM. Two independent experiments were performed with similar results (n = 2 per group).

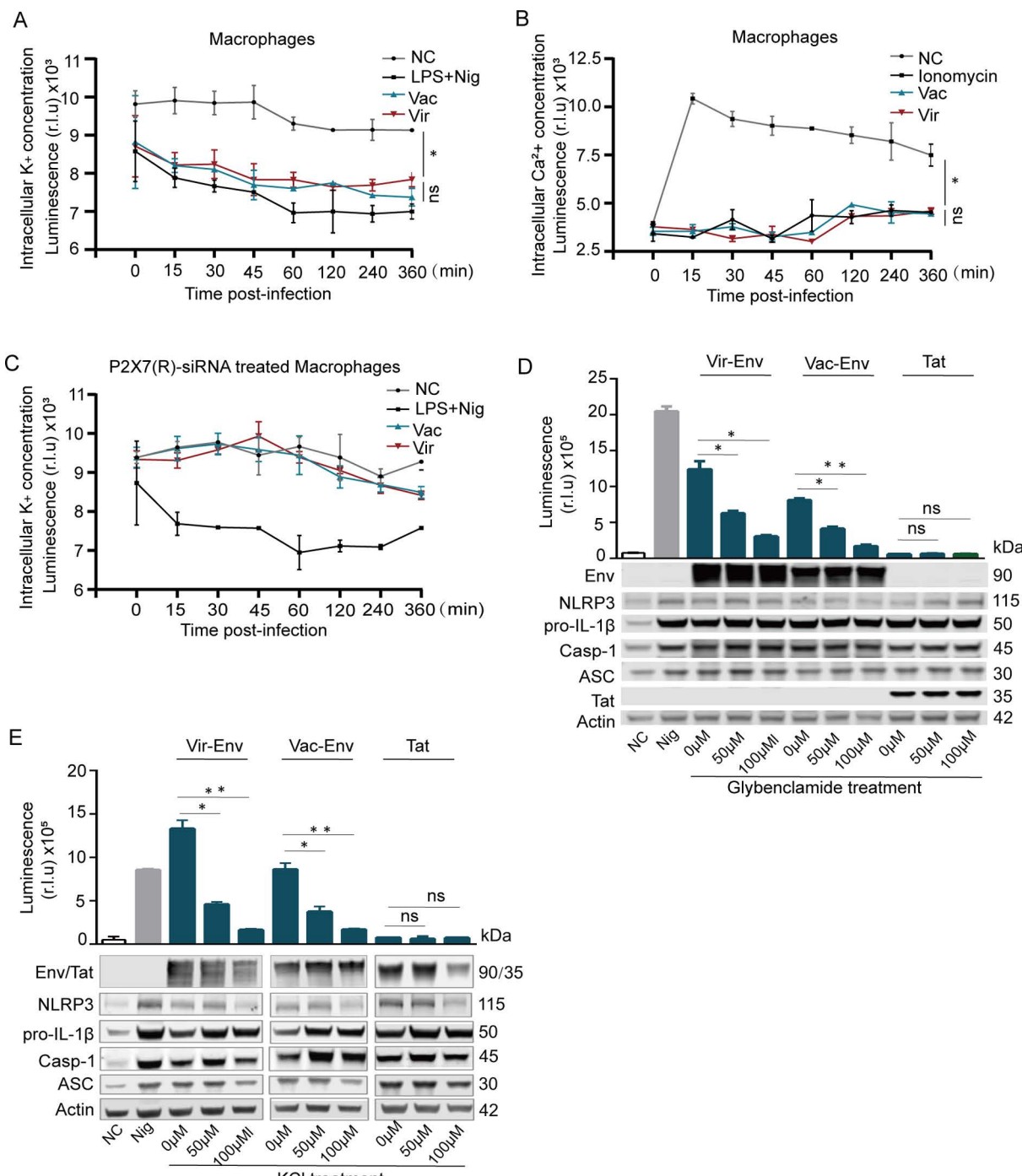

**Fig 4. Screening for ion-dependent P2X7 (R)-initiated NLRP3-IL-1β-axis activation induced by virulent-Env or by vaccine-Env. (A)** Time-dependent changes in intracellular K+ concentrations in macrophages infected with virulent EIAV or vaccine EIAV. Here NC refers to negative control and LPS+Nigericin (Nig) means positive control. **(B)** Time-dependent changes in intracellular Ca²⁺ concentrations in macrophages infected with virulent EIAV or vaccine EIAV. **(C)** Time-dependent changes in intracellular K+ concentrations in macrophages pre-treated with P2X7 (R) -specific siRNA for 6 h, and then infected with virulent EIAV or vaccine EIAV. **(D-E)** Evaluation IL-1β in supernatants of 293T cells co-transfected with virulent-*env* or vaccine-*env* in the presence of increasing doses of the K+ efflux inhibitor (Glybenclamide, 50 μM,100 μM) (D) and KCl (50 μM,100 μM) **(E)** (Tat served as a negative control) (* *P*<0.05, ** *P*<0.01). All data are mean of 2 independent experiments (n=2 per group).

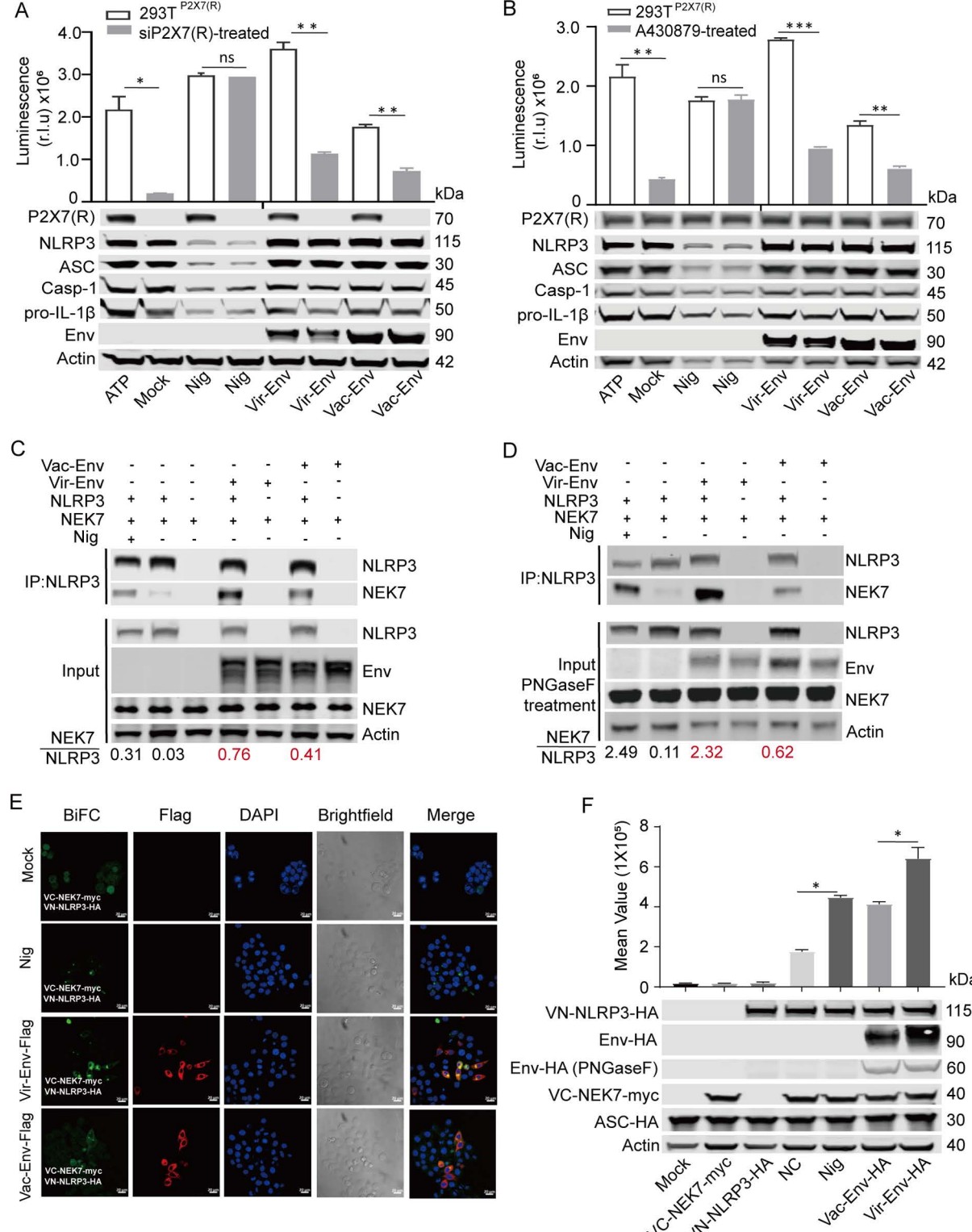

**Fig 5. Interaction between NLRP3 and NEK7 in 293T cells transfected with virulent-*env* or with vaccine-*env* .** (A) Evaluation the activation of the NLRP3-IL-1β axis in 293T^P2X7(R) cells transfected with control siRNA or P2X7 (R)-specific siRNA and then stimulated with virulent-*env* or vaccine-*env*. (B) Evaluation the activation of the NLRP3-IL-1β axis in 293T^P2X7(R) cells transfected with virulent-*env* or with vaccine-*env* in the absence or presence of

A430879 (a P2X7 (R) inhibitor). **(C)** Co-immunoprecipitation of HA-NEK7 or Flag-NLRP3 with HA-EIAV-*env* in 293T cells transfected with the indicated expression plasmids. **(D)** Same as (C) but samples were treated with PNGase F (Glycerol-free) for 1 h before immunoprecipitation. **(E)** Confocal experiments using a BiFC assay to assess the assembly of GFP-NLRP3 and HA-NEK7 in 293T cells co-transfected with virulent-*env* or vaccine-*env*. **(F)** Flow cytometry analysis was conducted alongside a BiFC assay to evaluate the assembly of VN-NLRP3-HA and VC-NEK7-myc in 293T cells co-transfected with virulent-*env* or vaccine-*env*. The mean fluorescence intensity (MFI) of the BiFC signals from transfected cells was normalized to the signals from un-transfected cells and presented as relative values (shown up). Western blot analyses confirmed the protein expression of the indicated plasmids in 293T cells (shown below). With the exception of F, which represents the outcome of three independent experiments (n = 3 per group), the remaining data are derived from two independent experiments (n = 2 per group).

simultaneously into 293T cells, a notable increase in NLPR3-NEK7 precipitation was observed in cells transfected with either vaccine-*env* or virulent-*env* compared to the mock-transfected group (Fig 5C). This observation was further confirmed through validation with PNGaseF treatment (Fig 5D). Subsequently, using Bimolecular Fluorescence Complementation (BiFC) analysis, we confirmed that both virulent-Env and vaccine-Env strengthen the NEK7-NLRP3 interaction, with virulent-Env showing a significantly stronger enhancing effect (Fig 5E and 5F). We further validated the formation of the NLPR3-NEK7 complex specifically within the Golgi apparatus and the endoplasmic reticulum, rather than in the mitochondrion (Figs 6A, 6B and S3), in 293T cells treated with Nigericin or transfected with either vaccine-*env* or virulent-*env*. Importantly, we observed that elevated $K^+$ concentrations had no effect on the interactions of Env with NLRP3 or NEK7 (S2 Fig). These findings suggest that compared to virulent-Env, vaccine-Env attenuates NLRP3-IL-1β-axis activation by weakening the association between NEK7 and NLRP3.

### Four amino acid residues in Env contribute to the differential regulation of the NLRP3-IL-1β-axis by the vaccine and virulent EIAV strains

We aimed to identify the differences between the virulent-Env and the vaccine-Env that were responsible for the apparent differential activation of the NLRP3-IL-1β-axis. To address this, we aligned the amino acid sequences of Env from virulent and vaccine strains of EIAV to identify any genetic variations. We found four single amino acid mutations (S235R, 236D-, N237K and N246K) are the primary conservative mutations during the attenuation process of the EIAV virulent strain (S4 Fig). Previous study has identified that mutations at these four specific sites exhibit a significant correlation with alterations in EIAV virulence [26,27]. Thus, we focused on these four sites to perform site-directed mutagenesis in *gp90*, resulting in the creation of two Env variants with reverse mutations from virulent Env and vaccine Env (Fig 7A and 7B). Remarkably, the mutant (vir-mut) successfully attenuated the virulent Env (vir-wt), aligning its activity closely with that of the Env (vac-wt). Conversely, this mutant (vac-mut) also enhanced the activity of the attenuated Env (vac-wt) to a level comparable to that of the Env (vir-wt) (Fig 7C). These results underscore the functional impact of specific mutations within Env on NLRP3-IL-1β-axis activation, shedding light on the intricate mechanisms underlying the differential immunomodulatory properties of vaccine-Env and virulent-Env. To further validate these findings, we generated libraries pseudotyped with vaccine-Env, virulent-Env, and Env mutants. Encouragingly, consistent results were observed in macrophages infected by these Env-pseudotyped viruses (Fig 7D). We next co-transfected three plasmids (*Env*, *NLPR3* and *NEK7*) into 293T cells, and validated the precipitation of NLPR3-NEK7 in the 293T cells. We found that the interaction between NEK7 and NLRP3 decreased upon Env (vir-mut) stimulation, compared with stimulation from Env (vir-wt). Conversely, a stronger interaction between NEK7 and NLRP3 was observed following treatment with vac-mut than with Env (vac-wt) (Fig 7E and 7F). Subsequently, we predicted the Env structures of the virulent and attenuated strains of EIAV using AlphaFold2. In the virulent strain, the residues at positions 235–237 and 246 are located within two protruding loop regions, forming a groove between them (Fig 8A). In contrast, mutations at R235 and K246 in the vaccine-Env result in the formation of a salt bridge, which occludes the groove (Fig 8A). Additionally, the deletion of D236, along with changes at these critical sites, alters the charge distribution and induces conformational changes in the groove (Fig 8B). As a result, the vaccine-Env loses the groove structure present in the virulent-Env. We speculated that this structural alteration in vaccine-Env may impair it

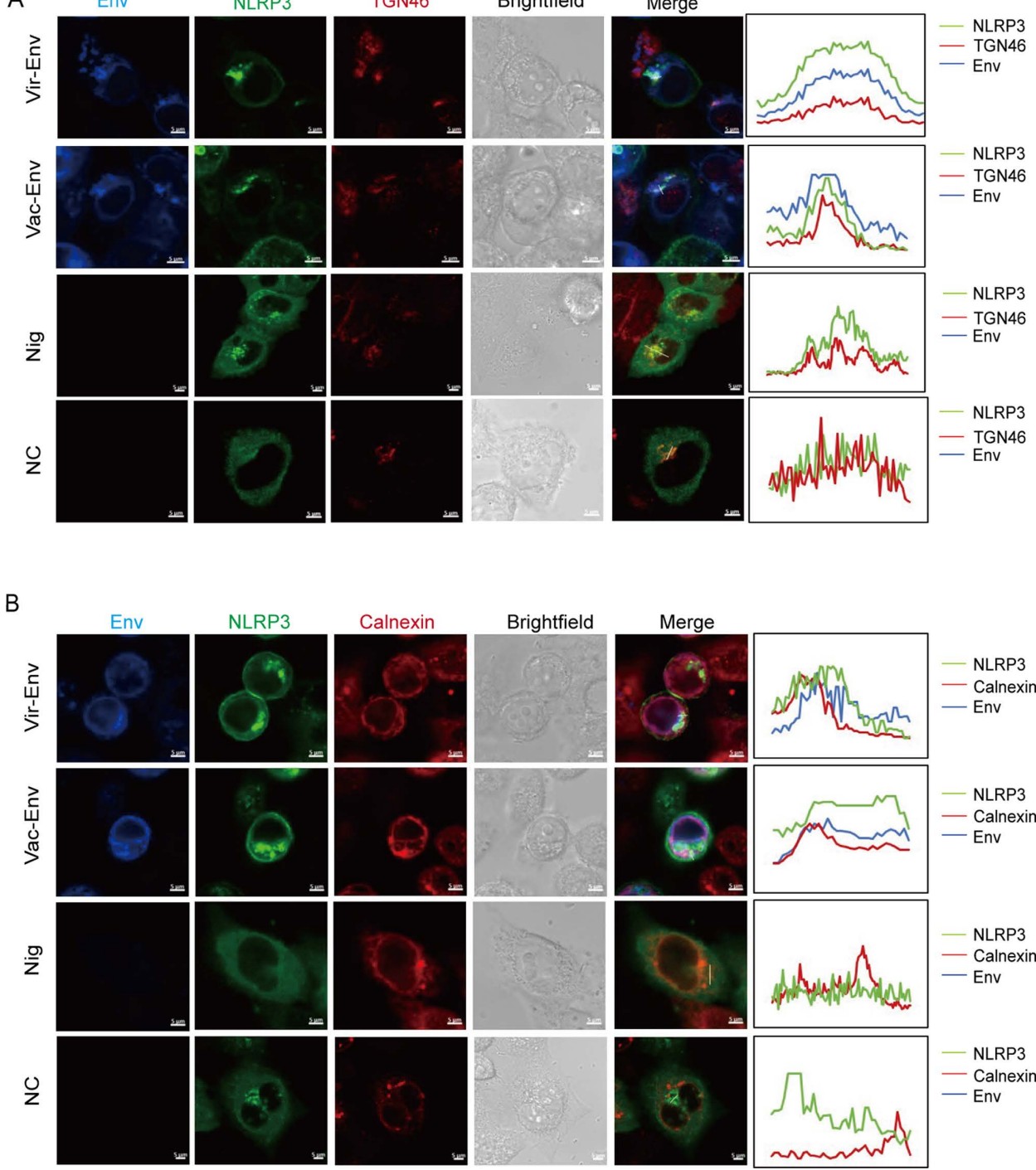

**Fig 6. Visualization of NLRP3 inflammasome assembly in 293T cells transfected with EIAV-*env* (s).** (**A-B**) Confocal micrograph of assembly of GFP-NLRP3 and HA-NEK7 in TGN46-labeled Golgi apparatus (A) and in Calnexin-labeled endoplasmic reticulum (B) in 293T cells co-transfected with virulent-*env* or vaccine-*env*.

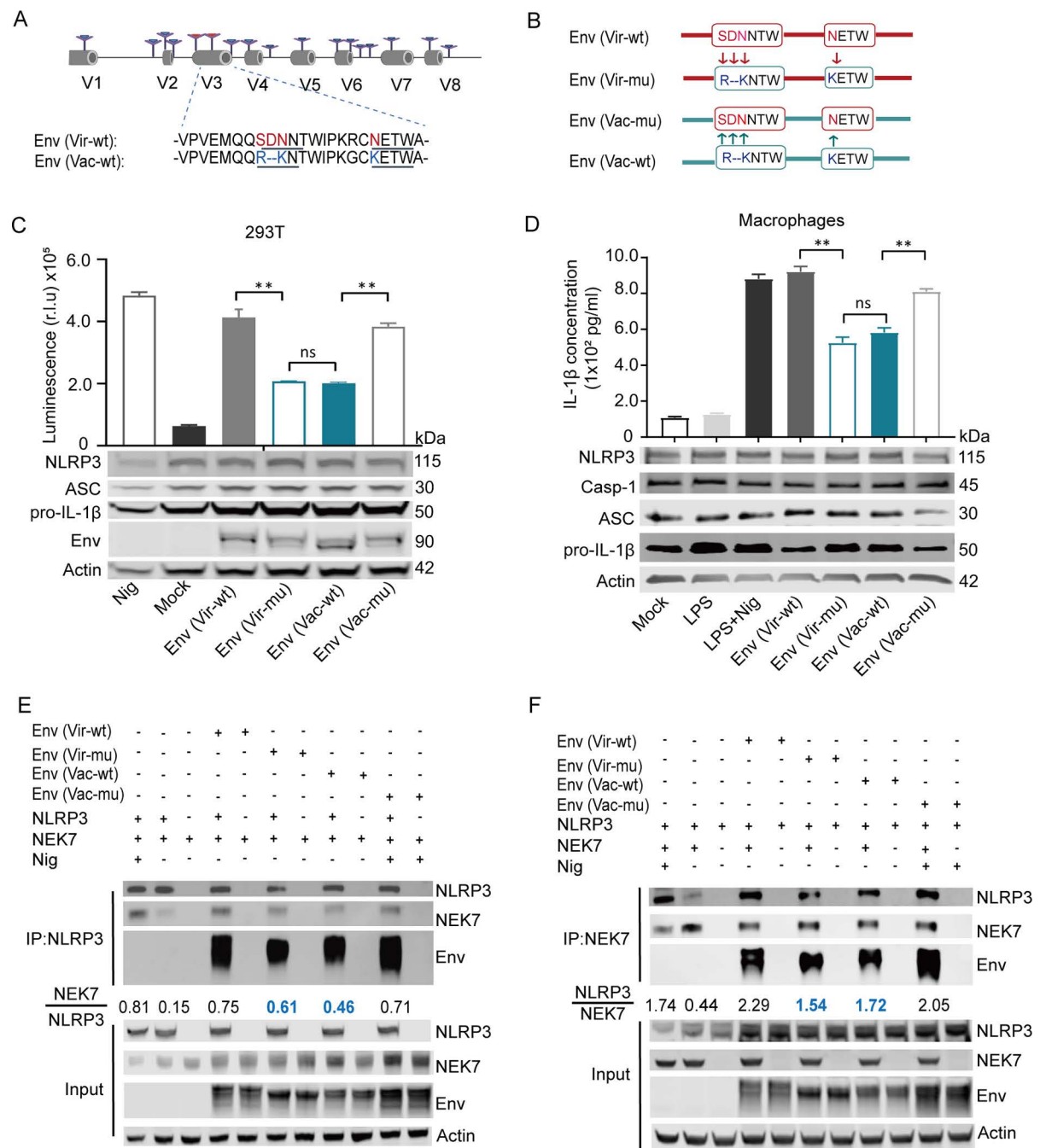

**Fig 7. Four amino acid residues are critical for the attenuation of the NLRP3-NEK7 complex assembly induced by vaccine EIAV. (A)** Schematic representation of the series of mutations constructed by substituting the indicated amino acids, which differ between virulent-Env (Gp90) and vaccine-Env (Gp90). Clip art created in BioRender Yuezhi L (2025). https://BioRender.com/g6u1ivt. **(B)** The amino acid sequences of virulent-*env* and vaccine-*env* are written in red and blue, respectively. **(C)** Statistical analyses from Western blot and luciferase activity assays demonstrate the expression levels of IL-1β in 293T cells transfected with EIAV strains (virulent-env, vaccine-env or one of their mutants) using the NLRP3 screening system.**(D)** Representative images and statistical analyses from Western blot and ELISA demonstrate the expression levels of IL-1β in equine macrophages treated with Env-pseudotyped viruses (vaccine-Env, virulent-Env and their mutants) or with LPS and Nigericin as a positive control.**(E-F)** The interactions of NEK7 and NLRP3 with virulent-Env, vaccine-Env or their indicated mutants were analyzed using Co-IP and reciprocal Co-IP assays. Two independent experiments were performed with similar results (n = 2 per group).

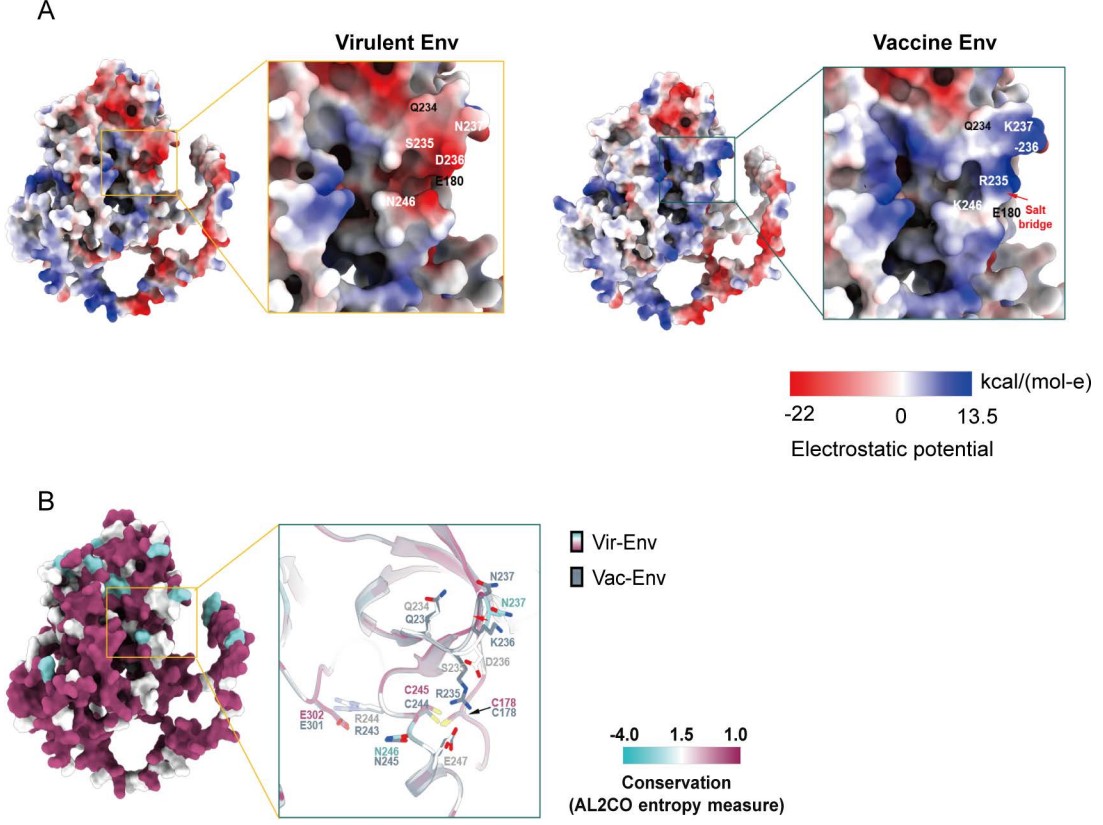

**Fig 8. Structural features of virulent-Env and vaccine-Env predicted model. (A)** The molecular surface of Gp90 models from virulent or vaccine strain are presented, colored by electrostatic potential. **(B)** A conservation analysis of Gp90 from virulent or vaccine strain were carried out using the Consurf tool within the UCSF ChimeraX 1.8 (https://www.rbvi.ucsf.edu/chimerax). The residues of Env (s) were color-coded according to their conservation grades, derived from sequence alignment results across virulent or vaccine strain in this study. Details of the key residues at of Gp90 are shown.

ability to bind to other proteins. These findings indicate that the mutations in the four identified amino acid residues, resulting in the structural alteration of the groove, could play a critical role in driving the differential NLRP3 activation outcomes between the virulent and vaccine strains.

## Discussion

The findings from your previous studies on horses inoculated with vaccine EIAV compared to those infected with the virulent strain are intriguing. It's fascinating that the vaccine strain led to reduced IL-1β levels and alleviated inflammatory pathologies [4,14]. As no assay *in vitro* to examine the NLRP3 inflammasome regulation by EIAV, we initially constructed a platform to systematically visualize the NLRP3-IL-1β axis in cells. The system was able to identify the molecular target of EIAV that responds to the host NLRP3-IL-1β axis. Using this platform, we first identified the molecular target of EIAV-Env that responded to the host NLRP3-IL-1β-axis. Next, we identified $K^+$ efflux associated with the Env-P2X7 (R) interaction. Interestingly, Env from either the virulent or the attenuated vaccine strain of EIAV was able to interact with P2X7 (R), and leading to only non-significant differences in $K^+$ ion efflux between the two groups. At the intracellular level, the formation of the NLRP3-NEK7 complex and subsequent pathway activation triggered by Env-NLRP3 and/or Env-NEK7 binding highlight intricate molecular interactions underlying the inflammasome pathway (Fig 9). The proposition that Env-NLRP3 and/or Env-NEK7 binding brings NLRP3 and NEK7 proteins into close proximity, thereby promoting the aggregation of

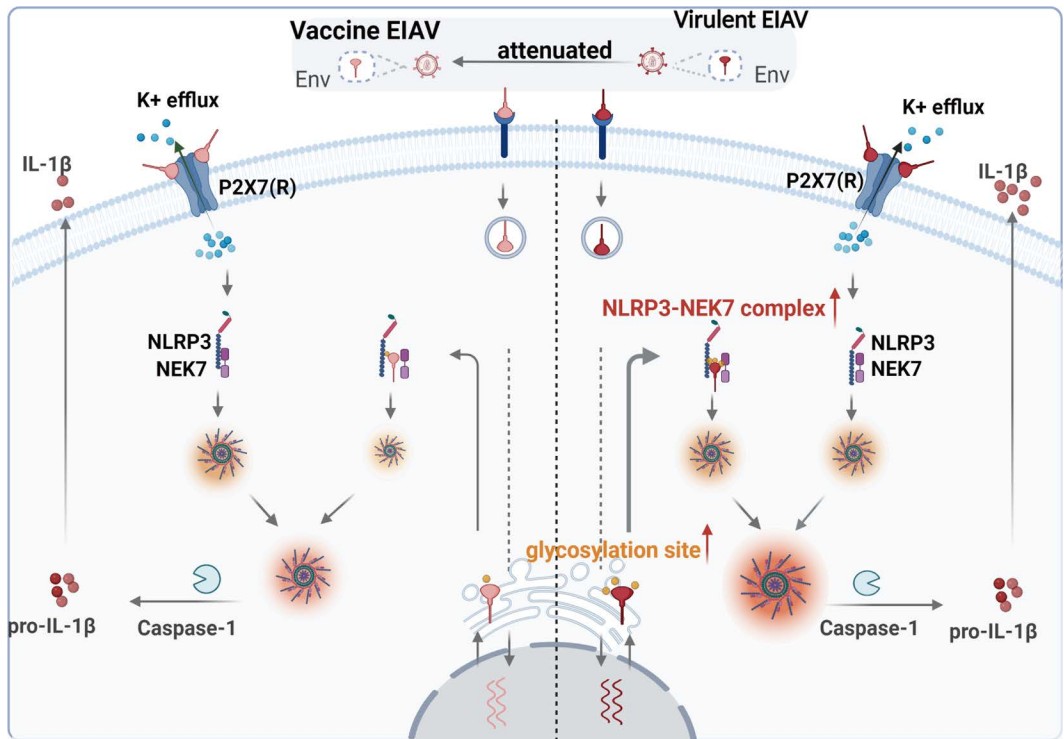

**Fig 9. Schematic representation of the mechanism underlying the attenuation NLRP3-IL-1β-axis induced by EIAV vaccine.** Both vaccine-Env and virulent-Env mediate NLRP3-IL-1β-axis activation via two-dependent steps. At the attachment stage of EIAV, Env (s) were able to directly bind to P2X7 (R) to induce K$^+$ efflux. There was no significant difference between two groups at the stage of the Env-P2X7 (R) interaction. After the virus enters the cells, expressed virulent-Env and vaccine-Env proteins directly bind to NEK7 and NLRP3 to promote the NEK7-NLRP3 interaction, leading to the assembly and activation of NLRP3 inflammasome. During this stage, vaccine-Env interacted only loosely and sparsely with NLRP3 and NEK7, attenuating NLRP3-IL-1β-axis activation compared to its virulent counterpart. Created in BioRender. Yuezhi, L. (2025) https://BioRender.com/l41b166.

the NLRP3-NEK7 complex in cells, provides valuable insight into the mechanistic aspects of EIAV-induced inflammation. The significant differences observed in the Env-mediated NLRP3-NEK7 complex between the vaccine and virulent EIAV strains, with vaccine-Env interacting weakly and sparsely with NLRP3 and NEK7, further emphasize the potential mechanisms underlying the differential modulation of NLRP3-dependent IL-1β activation by the two strains.

Systematic sequencing of the vaccine and virulent EIAV strains allowed us to identify an evolution-dependent adaption in the vaccine strain [8,13]. Such "natural" evolution-dependent adaption of an EIAV strain could theoretically be as a result of the delicate regulation of activation of the host inflammatory and immune responses, resulting in a decrease in pathological severity. Interestingly, although the amino acid sequences vary between vaccine-Env and virulent-Env, both Env proteins can still bind NLRP3 and NEK7 with similar affinity. However, the extent to which the formation of the NLRP3-NEK7 complex is triggered varies between these two viral strains, ultimately resulting in the differential activation of the NLRP3 inflammasome. Our preliminary studies have demonstrated a significant correlation between the mutation frequency at these four sites and changes in viral virulence, along with the loss of two N-glycosylation sites in the vaccine strain. To elucidate the molecular mechanisms underlying this phenomenon, we focused on four key conserved mutation sites during the attenuation process of the EIAV virulent strain. Site-directed mutagenesis of these four critical residues in the vaccine-Env to mirror the corresponding virulent-Env markedly potentiated NLRP3-IL-1β axis activation, achieving levels equivalent to those elicited by the virulent strain. Conversely, reverting these mutations in the virulent-Env to match the vaccine-Env substantially attenuated NLRP3-IL-1β axis activation. Structural prediction analysis further reveal that

four key mutations cause the vaccine-Env to form a salt bridge, altering the surface charge distribution. These changes induce conformational shifts in the loop region of the corresponding domain in the virulent-Env, thereby reducing the ability of the vaccine-Env to bind with NLRP3 and NEK7. We propose that the key amino acid mutations occurring during the attenuation of the virulent strain leads to conformational changes, which result in differential binding to NLRP3 and NEK7. This variation in binding is the primary factor behind the strain-specific regulation of NLRP3 activation.

Our team has demonstrated that the high diversity in the Env of the EIAV vaccine strain leads to greater protective immunity against viral pathogenesis both *in vitro* and *in vivo* [13,22,26]. This observation aligns with similar findings reported by other researchers [6,28,29], further supporting the stress-dependent flexibility of the immune response against lentiviruses. This study revealed conserved mutations occurring during the attenuation of the EIAV virulent strain, which resulted in differential regulation of the NLRP3 inflammasome by the virulent and attenuated EIAV strains. These findings further underscore the evolutionary interplay between the virus and its host. Therefore, the intricate regulation observed in the vaccine-induced response likely reflects the outcome of a finely tuned evolutionary interplay between the virus and its host [30–32].

In conclusion, the EIAV attenuated vaccine exhibits a representative stable and finely tuned variation, which enables precise and adaptable activation of the NLRP3-IL-1β axis through both Env-P2X7 (R)-mediated transcellular signaling and Env-NLRP3-NEK7-mediated intracellular signaling. Such unique proximity-dependent interaction might benefit a "win-win" adaption during EIAV infection. Our evolution-based insights could provide an alternative strategy for the development of envelope-based vaccines.

## Materials and methods

### Ethics statement

The horses used in this study were approved by the Harbin Veterinary Research Institute (HVRI), the Chinese Academy of Agricultural Sciences (CAAS). All physical procedures associated with this work were done under anesthesia to minimize pain and distress in accordance with the recommendations of the Ethics Committees of HVRI. The Animal Ethics Committee approval number is Heilongjiang-SYXK (Hei) 2017-009.

### Viruses and cells

Two EIAV strains were used in our study: i) $EIAV_{DLV34}$ was derived from a cell culture-adapted EIAV virulent strain, which resulted in acute EIA in all the experimentally infected horses at $2 \times 10^5$ $TCID_{50}$. ii) The attenuated EIAV vaccine strain ($EIAV_{DLV121}$) was developed from a natural virulent EIAV strain by successive passages in donkey monocyte-derived macrophages (dMDMs), and is here termed the vaccine strain. Equine monocyte-derived macrophages and 293T cells were used for infection and transfection separately. Equine monocyte-derived macrophages (eMDMs) were enriched from whole blood as described previously [3,4]. 293T cells (ATCC, CRL3216) were cultured on DMEM (Sigma-Aldrich, D0819) containing 10% (vol/vol) fetal bovine serum (FBS; Wisent, 086–15) and antibiotics (1% penicillin/streptomycin, Thermo Fisher Scientific, 15,070,063).

### Reagents and antibodies

The following antibodies were used in this study: mouse anti-equine NLRP3 mAb, rabbit anti-equine Caspase-1 pAb, rabbit anti-equine ASC pAb, rabbit anti-equine IL-1β pAb, mouse anti-equine IL-1β mAb and mouse anti-Env mAb were prepared by our laboratory. MCC950 sodium and A430879 were purchased from MCE (HY-12815 and HY15488, USA). Glybenclamide (Gly) was purchased from Novus Biologicals (NBP2-30141, USA). Lipopolysaccharide (LPS) (L2280) and ATP (A7699) were purchased from Sigma-Aldrich (St. Louis, MO, USA). TGN46, Tom20 and Calnexin were purchased from Abcam (ab174280, ab186735 and ab133615, USA). Nigericin was obtained from Sigma (28643-80-3, USA). NEK7

antibody (1:1000) was purchased from Abcam. Antibodies against Flag (F3165) (1:2000), rabbit anti-HA (H6908) (1:2000), mouse anti-myc (05-724MG) (1:4000) and monoclonal mouse anti-Actin (G9295) (1:5000) were purchased from Sigma (St Louis, USA). Secondary antibodies included DyLight 680 Labeled Anti-Rabbit IgG Antibody (35568, Invitrogen); DyLight 800 Labeled Anti-Rabbit IgG Antibody (072-06-15-16, KPL); Goat Fluor 488 labeled anti-Rabbit IgG (A-11034, Invitrogen); Alexa Fluor 647 labeled Goat anti-Rabbit IgG (A-2124, Invitrogen); Goat anti-Mouse IgG cross-adsorbed secondary antibody (A-1101, Invitrogen). Lipofectamine 2000 and RNAiMAX transfection reagent were purchased from Invitrogen Corporation (11668019 and 13778075, USA). The equine IL-1β ELISA kit was obtained from the R&D system (DY3340, USA). The IPG-SPEC Assay Kit was purchased from ION Biosciences (3011F, USA). Fluo-4 AM (Calcium Ion Fluorescent Probe, 5mM) was purchased form Beyotime (S1056).

## Transfection and infection

For transfection, 293T cells were seeded into 6-well plates (2 x $10^5$ cells/well) and singly or co-transfected with siRNA oligos (50nM) using RNAiMAX transfection reagent (Invitrogen) or the indicated plasmids DNA using Lipofectamine 2000 (Invitrogen) according to the manufacturer's instructions. The expression of these target proteins was confirmed using western blotting 24 hours after transfection. For virus infection, eMDMs were enriched from whole blood and were cultured as previously reported [3]. The eMDMs (1 x $10^5$ cells/well) were cultured for one day and then infected with EIAV strains at 2 x $10^5$ $TCID_{50}$. For the positive control group, cells were first treated with LPS for 6 hours, followed by Nigericin stimulation for 2 hours prior to sample collection. Cells and supernatants were collected at the indicated time-points. The expression of the inflammasome components was confirmed using western blotting. Supernatants were collected for analysis of mature IL-1β using an ELISA kit (R&B, USA).

## Histopathology and pathological assessment

For the histopathology studies, three horses were inoculated with attenuated vaccine and three horses were infected with virulent strain as previously described [3,22]. Thirty days after inoculation, the horses were euthanized and the organs were collected and examined. Liver, lung, spleen, kidney and lymph node of each horse were embedded in paraffin for 24h, then sliced and stained with haematoxylin and eosin (H&E). Semi-quantitative analysis of inflammatory pathological changes in each organ was scored independently as previously described [22]. Pathological assessments of five autopsied organs were performed by three independent professional pathologists in a double-blind experiment.

## Assay for NLRP3-IL-1β-axis activation and IL-1β measurement

The NLRP3-IL-1β-axis screening system, which monitors the cleavage of pro-IL-1β, was developed to study equine NLRP3 inflammasome activation based on the technique published in a previous report [33]. This reporter is based on the biological activity of a pro-IL-1β-*Gaussia* luciferase (iGluc) fusion. Briefly, the pro-IL-1β-dependent formation of a protein complex renders the iGluc enzyme inactive. The cleavage of pro-IL-1β leads to monomerization of this biosensor protein, and the fusion protein then gains strong luciferase activity. According to the working principle of this system, expression plasmids (pCDNA3.1) coding for eqCaspase-1, eqASC, eqNLRP3 and eqIL-1β were constructed using the INFUSION cloning technique (Clontech). 293T cells (2 x $10^5$ cells/well) were then transiently transfected with the inflammasome plasmid constructs with a total of 463.75ng DNA per 12-well plate (pCDNA3-eqASC-HA 3,75ng, pCDNA3.1-eqCaspase1-HA 3.75ng, pCDNA3-eqNLRP3-HA 6.25ng, pCDNA3.1-pro-IL-1β-*Gaussia* 250ng, other selected plasmid 200ng). Two hours before sample collection, cells of positive control group were stimulated with 10 μM Nigericin or ATP (1mM). At 24h post-transfection, the supernatant of the cultured cells was measured to assess the luciferase signal using Coelenterazine (55779-48-1, Sigma). Adherent cells were lysed in passive lysis buffer (E1941, Promega) so that the western blot assay could be run in parallel. IL-1β measurement was carried out with an equine IL-1β ELISA kit (R&B, USA).

## Real-time RT-PCR

Total RNA was extracted using TRizol (Invitrogen) according to the supplier's instructions. Real-time RT-PCR was performed using the SYBR-Green method as described previously using available primers. cDNA was amplified for the IL-1β gene (sense: 5'-GAGGAGGATGGCCCAAAACA-3'; anti-sense: 5'-AGC CACAATGATTGACACGAC-3'), the *NLRP3* gene (sense: 5'-TCGGTTGGTGAACTG CTCTC-3'; anti-sense: 5'-CGGTGAGGCTCCAGTTAGTG-3'), the *NLRC4* gene (sense: 5'-AAGTCTCTGTCAGCCGAACC-3'; anti-sense: 5'-CGAGACTGCTCTCCTTCAGT-3'), and the *NLRP1* gene (sense: 5'-CACACCTGGAAAGGAATCAGAGA-3'; anti-sense: 5'-TCCGGGGTCAGATGTGTGTA-3'). Actin mRNA (sense: 5'-CATCTGCTG GAAGGTGGACAA-3'; anti-sense: 5'-CGACATCCGTAAGGACCGTTA-3') was examined as the cellular reference control. The relative expression of each gene was calculated using delta-delta-Ct method ($2^{-\Delta\Delta Ct}$).

## Western blots and immunoprecipitation

At 24 h after transfection as described above, cells were lysed in an appropriate cell lysis buffer (1% Triton X-100, 50 mM Tris-HCl, pH 7.4, and 150 mM NaCl) with 1% Protease Inhibitor Cocktail (HY-K0010, MCE). After running the lysates on a 4%-12% or 12% SDS-PAGE gel, proteins were transferred onto a PVDF membrane. The blots were probed with primary antibodies at RT for 1–2 hours and then incubated with the indicated secondary antibodies (1:10000). For immunoprecipitations, cell pellets were lysed and incubated overnight with the appropriate antibodies plus anti-Flag (M8823, Sigma-Aldrich) or anti-HA magnetic beads (HY-K0201, MCE) as described previously. Signals were detected using a Licor Odyssey 448 imaging system (USA). For relative protein expression, images were quantified in ImageJ software and normalized using the corresponding endogenous β-actin expression.

## Confocal microscopy

For immunostaining, 293T cells or primary eMDMs were seeded on coverslips and transfected (or infected) with the indicated plasmids (or EIAV strain). After 24 h post-transfection, or following infection at 6 h, 12 h and 24 h, cells were fixed in 4% paraformaldehyde (P0099, Biotime) for 15 min and subjected to saponin permeabilization (P0050, Biotime). Cells were then incubated with the corresponding antibodies and suitable secondary antibodies. Nuclei were stained with DAPI (P0131, Beyotime) for 10 min. All cells were washed with PBS 5 times after each step. Fluorescence images were acquired using a confocal laser scanning microscope (Carl Zeiss LSM 800), and analyzed using ZEN 2 (blue edition) software.

## Mass spectrometry

The eMDMs were separated and cultured as described previously [4], and then were infected with an EIAV strain. Three days post-infection, cells were collected by scraping into a lysis buffer (1% Triton X-100, 50 mM Tris-HCl, pH 7.4, and 150 mM NaCl). The cell lysates were cleared using centrifugation (15,000 r.p.m.,10 min). Dilution buffer and then normal mouse IgG2b and anti-Env magnetic beads (HY-K0202, MCE) were added into the supernatant, and incubated for 2 h at 4 °C with rotation. The supernatant was removed, and the beads were washed extensively with 1 x PBST. An epitope peptide targeting Env was then added (1mg/ml) to disrupt the interaction between Env and the beads. The eluted samples were loaded on a large 12% SDS-PAGE (4561068, BioRad) and resolved, and were then analyzed using western blot. Electrophoresis was run for longer (40–50 min) than usual to ensure good band resolution. The stained protein bands were then cut out and incubated in deionized water for further mass spectrometric analysis. Mass spectrometry was performed by the BGI Genomics company. Briefly, each sample was dehydrated, reduced, alkylated, and then digested with trypsin. The peptides were extracted and delivered onto a nano RP column and eluted with a gradient. Peptide identification was carried out in the Mascot software (Version 2.3.01, Matrix Science, UK) using the UniProt database search algorithm and the integrated false discovery rate (FDR) analysis function. The data were searched against a protein sequence database downloaded from 2019 uni_horse horse (50,340 sequences; 30,790,782 residues).

## EIAV-Env pseudovirus construction

Briefly, the packaging constructs psPAX2 (AIDS Resource and Reagent Program), luciferase reporter plasmid Plenti-CMV Puro-Luc (Addgene), and the plasmids expressing the virulent-Env, vaccine-Env or Env mutants were used for EIAV-Env pseudovirus generation. These plasmids were co-transfected into 293T cells ($5 \times 10^4$ cells/well) using calcium phosphate. After 48h transfection, supernatants were collected and infectivity was determined with reverse transcriptase activity assays (11468120910, Sigma).

## RNAi

293T cells were transfected with a pool of 3 P2X7 (R)-targeting siRNAs (TGACAGAAATTGACAACAA; AGACAAGAACAACTCCAAA; ACAGTGTCTTAACATTCAA) or control siRNA (CAAACAGAAUGGUCCUAAA) (50 µM) using RNAiMAX transfection reagent (13778100, Invitrogen) according to the manufacturer's instructions. The efficiency of siRNA silencing of P2X7 (R) was evaluated using western blotting.

## Generation of the stable P2X7 (R) 293T cell line

Briefly, the pSIN4-P2X7 lentivirus was generated by co-transfecting 293T cells with packaging plasmids psPAX2 and pMD2.G. Medium containing viruses was filtered and added to the target cells. 24h following initial infection with the lentiviruses, cells were selected on puromycin (1µg/µl) (A1113803, Gibco) for at least 14 days. The remaining cells were enriched using fluorescence-activated cell sorting (FACS) and were then expanded until there were enough cells for validation with immunoblotting.

## Bimolecular Fluorescence Complementation assay

For BiFC imaging, 293T cells were seeded on glass-bottom dishes and co-transfected with pCDNA3.1-VN-eqNLRP3-HA, pCDNA3.1-VC-NEK7-myc, pCDNA3.1-ASC-HA and pCDNA3.1-vir-*gp90*-Flag or pCDNA3.1-vac-*gp90*-Flag vectors. After 24h, cells were fixed with 4% PFA, permeabilized with 0.1% saponin, and blocked with 5% BSA. Primary antibodies were applied overnight at 4°C, followed by Alexa Fluor 405/647-conjugated secondary antibodies for 1h at RT in the dark. Cells were washed with PBS, stained with DAPI, and imaged using a Zeiss LSM 800 confocal microscope with ZEN 2.3 software. For BiFC flow cytometry, 293T cells were seeded in 24-well plates and co-transfected with the plasmids described above. Two hours before sample collection, cells of positive control group were stimulated with 10 µM Nigericin. After 24h, cells were trypsinized, pelleted, washed twice with ice-cold PBS, and resuspended in 1000 µl PBS. Suspensions of $1 \times 10^5$ cells were analyzed for BiFC fluorescence MFI.

## Measurement of intracellular potassium

Intracellular $K^+$ concentrations were determined with a potassium probe (IPG-2AM) according to the manufacturer's instructions. Briefly, equine macrophages were cultured and plated in a 96-well microplate ($1.5 \times 10^5$ cells/well) as described previously [4]. On the day of the experiment, equine macrophages were first infected with either EIAV virulent or vaccine strains separately at the same $TCID_{50}$, and were then either treated with Nigericin (10 µM) (28643-80-3, Sigma) at the indicated time-points (0 min, 15 min, 30 min, 45 min, 60 min, 120 min, 240 min and 360 min), or left untreated. 100 µl/well potassium probe was then added. After 1h incubation at 37 °C, fluorescent signals were detected using an ELISA plate reader, with excitation/emission wavelengths of 520/540 nM. All reagents in this assay were provided in the IPG-SPEC Assay Kit (3011F, ION Biosciences).

## Measurement of intracellular $Ca^{2+}$

Intracellular $Ca^{2+}$ concentrations were measured in equine macrophages using the $Ca^{2+}$ sensitive dye, Fluo-4-AM. Briefly, equine macrophages were infected with either EIAV virulent or vaccine strains at the same dose. At the indicated

timepoints, cells were switched and washed three times with wash buffer and incubated with 10 µM Fluo-4-AM for 45 min at 37 °C, followed by three washes and an additional incubation in PBS for 20 min. The fluorescence value (excitation at 488 nm) was monitored using a laser scanner, and the emitted light signal was read at 510 nm.

### Protein structure prediction

To analyze the structure features of key residues from Env (Gp90) in the virulent or vaccine strain, we utilized the Alpha-Fold 2 server (https://colab.research.google.com/github/sokrypton/ColabFold/blob/main/AlphaFold2.ipynb) to predict two structural models. Full length amino acid sequences of Gp90 from virulent and vaccine strains were used. By default, AlphaFold 2 generates five predicted structures from a single seed by sampling the diffusion process five times. In this study, we selected the first model for further analysis. The model analysis was subsequently examined through visual inspection using UCSF ChimeraX (version 1.8) (accessible at https://www.cgl.ucsf.edu/chimera/) [34].

### Statistical analysis

Image J software was used to measure the band intensity from the western blot. GraphPad Prism was used for statistical tests. All results are expressed as mean ± SD, and statistical significance was analyzed using two-tailed unpaired t-test, two-way ANOVA or one-way ANOVA followed by Dunnett's multiple comparisons tests. (Prism, GraphPad). Error bars represent SD (standard deviation). ns, not significant ($P > 0.05$), $*P < 0.05$, $**P < 0.01$, $***P < 0.001$.

### Supporting information

**S1 Fig. Comparison of inflammatory pathological changes in specimens from horses inoculated with the EIAV vaccine and horses infected with the virulent EIAV strain.** Typical inflammatory pathological changes observed in lung, kidney, liver, spleen and lymph gland on infection with EIAV vaccine or virulent strains are presented separately (haematoxilin and eosin 4 m paraffin sections, original magnification 10x). Severity of pathological lesions and scoring are given in Table 1.
(TIF)

**S2 Fig. $K^+$ efflux affects the interaction between NEK7 and NLRP3, but not that between Env and NLRP3 or NEK7.** (**A**) 293T cells were co-transfected with NEK7-Flag and NLRP3-HA in the presence of 50 µM concentrations of KCl (increasing intracellular $K^+$ concentration). Immunoprecipitation and analyses were as described in Fig 3D. (**B-C**) Procedure was as in (A) but instead of transfection with NEK7 and NLRP3, cells were co-transfected with env-HA and NLRP3-Flag (B) or co-transfected with env-HA and NEK7-Flag (C) at high concentrations of KCl (50 µM). An immunoprecipitation assay was then performed using the indicated antibodies.
(TIF)

**S3 Fig. Visualization of NLRP3 inflammasome localization in 293T cells transfected with EIAV-env (s).** Confocal micrograph of assembly of GFP-NLRP3 and HA-NEK7 not Tom20-labeled mitochondria in 293T cells co-transfected with virulent-env or vaccine-env.
(TIF)

**S4 Fig. Sequence alignment of Env (Gp90) at critical passages during the attenuation process of the EIAV virulent strain.** Their conserved mutation sites between the virulent and attenuated vaccine strains were highlighted.
(TIF)

**S1 Data. Excel spreadsheet containing, in separate sheets, the underlying numerical data and statistical analysis for Figures.**
(XLSX)

## Acknowledgments

We would like to thank Professor Xin Yin for helpful discussions. We are grateful to Professor Yingying Guo for her invaluable contributions to protein structure prediction and analysis. We thank the Core Facility of the Harbin Veterinary Research Institute, the Chinese Academy of Agricultural Sciences for providing the technology platform.

## Author contributions

**Conceptualization:** Yuezhi Lin, Xiaojun Wang.

**Data curation:** Xing Guo, Cong Liu.

**Formal analysis:** Xing Guo, Yuhong Wang, Hongxin Li, Lei Na.

**Funding acquisition:** Yuezhi Lin, Xiaojun Wang.

**Investigation:** Xing Guo, Cong Liu, Hongxin Li, Saiwen Ma, Lei Na.

**Methodology:** Saiwen Ma.

**Project administration:** Yuezhi Lin, Xiaojun Wang.

**Software:** Yuhong Wang, Hongxin Li, Saiwen Ma.

**Validation:** Xing Guo, Cong Liu, Lei Na, Huiling Ren.

**Writing – original draft:** Xing Guo, Yuezhi Lin.

**Writing – review & editing:** Yuezhi Lin, Xiaojun Wang.

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
