## [Decision Letter · Decision Letter 0]

PPATHOGENS-D-24-02558

Env from EIAV vaccine delicately regulates NLRP3 activation via attenuating NLRP3-NEK7 interaction

PLOS Pathogens

Dear Dr. Wang,

Thank you for submitting your manuscript to PLOS Pathogens. After careful consideration, we feel that it has merit but does not fully meet PLOS Pathogens's publication criteria as it currently stands. Therefore, we invite you to submit a revised version of the manuscript that addresses the points raised during the review process. Note that we may send your paper back to some of the more critical reviewers upon resubmission.

Please pay particular attention to the following reviewer suggestions and give them due consideration:

Figure 1F shows only one EIAV and not both the virulent and attenuated viruses. Please complete this figure.

Figure 5E includes flow cytometry data to demonstrate complex formation between proteins. Please explain why this was used and if this method is appropriate to support the conclusion. 

Please explain how and why the 4 amino acids that are different between the virulent and attenuated virus were selected, how they contribute to virulence, and if additional differences exist. 

Please submit your revised manuscript within 60 days . If you will need more time than this to complete your revisions, please reply to this message or contact the journal office at plospathogens@plos.org. Please include the following items when submitting your revised manuscript:

We look forward to receiving your revised manuscript.

Kind regards,

Ulf Dittmer

Academic Editor

PLOS Pathogens

Susan Ross

Section Editor

PLOS Pathogens

Sumita Bhaduri-McIntosh

Editor-in-Chief

PLOS Pathogens

orcid.org/0000-0003-2946-9497

 Michael Malim

Editor-in-Chief

PLOS Pathogens

orcid.org/0000-0002-7699-2064

**Journal Requirements:**

At this stage, the following Authors/Authors require contributions: xing Guo, cong Liu, yuhong Wang, hongxin Li, saiwen Ma, lei Na, huiling Ren, yuezhi Lin, and Xiaojun Wang. Please ensure that the full contributions of each author are acknowledged in the "Add/Edit/Remove Authors" section of our submission form.

2) Please ensure that all Figure files have corresponding citations and legends within the manuscript. Currently, Figure Figure 8 in your submission file inventory does not have an in-text citation. If the figure is no longer to be included as part of the submission, please remove it from the file inventory.

3) We have noticed that you have uploaded Supporting Information files, but you have not included a complete list of legends. Please add a full list of legends for your Supporting Information files after the references list.

Potential Copyright Issues:

- Figure 3. Please confirm whether you drew the images / clip-art within the figure panels by hand. If you did not draw the images, please provide (a) a link to the source of the images or icons and their license / terms of use; or (b) written permission from the copyright holder to publish the images or icons under our CC BY 4.0 license. Alternatively, you may replace the images with open source alternatives. See these open source resources you may use to replace images / clip-art:

5) Please amend your detailed Financial Disclosure statement. This is published with the article. It must therefore be completed in full sentences and contain the exact wording you wish to be published. State what role the funders took in the study. If the funders had no role in your study, please state: "The funders had no role in study design, data collection and analysis, decision to publish, or preparation of the manuscript.".

**Reviewers' Comments:**

Reviewer's Responses to Questions

**Part I - Summary**

Reviewer #1: In their current manuscript, Guo et al. provide mechanistic insight into the differential induction of IL1beta by the wild-type and vaccine EIAV strains. They show that EIAV Env is the main EIAV protein responsible for the differential induction of IL1beta and use a range of methods to show that EIAV Env interacts with the proteins P2X7(R), NLRP3 and NEK7, which are components of the NLRP-IL1beta pathway. The authors finally identify 4 amino acid sequences that seem to be responsible for the different IL1beta induction by virulent and attenuated EIAV. The experiments are well conducted and well controlled and the data is convincing. There are a few issues that the authors should address:

Reviewer #2: Equine infectious anemia virus (EIAV) belongs to the lentivirus family, and this study demonstrated to us that Env from EIAV vaccine delicately regulates NLRP3 activation via attenuating NLRP3-NEK7 interaction, which is very interesting and offers more information on EIAV. However, there are some obvious weaknesses in this study.

1. authors try to display the differences of inflammatory respons and host responses between the attenuated vaccine and its virulent mother strain. However, the results seems to show that the virulence play important role in activate the NLRP3-IL-1β. The EIAV-Env of the attenuated vaccine or its virulent mother strain looks like to play the similar role in this study.

2.Env was the main factor affecting the effects of the NLRP3-IL-1β-axis activation induced by EIAV infection. Interestingly, no disparity between the abilities of Env to bind these proteins was observed between the virulent and vaccine EIAV strains . Notably, no differences were observed in the Env-mediated K+ efflux between the two EIAV strains. At the end, authors conclude that four amino acid residues in Env contribute to the differential regulation of the NLRP3-IL-1β-axis. Authors didnot discuss whether these amino acid residues influence the virulence or not.

**Part II – Major Issues: Key Experiments Required for Acceptance**

Reviewer #1: The authors state in line 103-104 that the experiment in Fig. 1F confirms that both virulent and attenuated EIAV activate the NLRP3-IL1beta axis , but the figure shows only one EIAV and it is not clear which one is shown. Please provide the additional data.

It is not at all clear to me how the flow cytometry data in Figure 5E can demonstrate the complex formation in contrast to cells just being (more or less) positive for one or both fluorescent labels. Please provide additional data supporting this claim, or provide more quantified data from confocal microscopy, which seems to be the more appropriate method.

Reviewer #2: Authors should display whether four amino acid residues in Env contribute to the virulence.

**Part III – Minor Issues: Editorial and Data Presentation Modifications**

Reviewer #1: The figures show a lot of bar charts and it is not clear how many data points were collected, and in how many independent experiments. Please provide this information in all figure legends.

line 122: please add this statistical difference to the figure

line 154: please avoid the abbreviation "Gly", you only use it two times in the text anyway and it makes it harder to understand for the reader

line 193: the authors mention "four key amino acids", they should elaborate whether these are the only different amino acids in the two EIAV Env sequences, or whether they picked them because of some special characteristics? in line 263, the authors mention a high diversity in Env of the EIAV vaccine strain, which implies that more than these 4 amino acids are different. Showin an alignment of the whole Env gp90 sequences might be helpful.

Furthermore, it would help the reader if the authors could provide insght about the location of these four amino acids in the 3D structure of the Env protein, ideally in a figure/schematic.

line 254-256: "N246 mutation in wt" -> please rephrase so that it is easier to understand which sequence carries which amino acid

line 271-273: the authors need to provide more context for the statement that EIAV "evolves flexibly rather than lethally", it is not clear what they mean.

To make the fluorescene images easier to interpret, please add brightfield images.

Figure 8: it is not easy to recognize the mechanism in this figure, the authors may want to change it to make the important differences easier to identify.

Plese avoid the speculation that a similar approach would lead to an HIV vaccine, this is a very big stretch.

Methods section: please make sure that enough details are provided so that results can be replicated, e.g. how much plasmid DNA was used for transfections, how many cells were infected with 2x10^5 TCID50 (->MOI)?

lines 464-466: the authors only mention labeled secondary antibodies, what primary antibodies were used?

line 472: the authors used a Student's t test for the statistical analysis, but this is only appropriate for the comparison of two groups. If comparing more groups, use either an ANOVA or perform an adjustment for multiple comparisons.

line 473: the authors state that the error bars in the figures show the "SD (standard error of the mean)". SD = standard deviation and SEM = standard error of the mean are not the same and not interchangeable, they should use the standard deviation and modify the figures accordingly.

Reviewer #2: No.

PLOS authors have the option to publish the peer review history of their article (what does this mean? ). If published, this will include your full peer review and any attached files.

**Do you want your identity to be public for this peer review?** For information about this choice, including consent withdrawal, please see our Privacy Policy .

Reviewer #1: No

Reviewer #2: No

**Figure resubmission:**
---

## [Decision Letter · Decision Letter 1]

PPATHOGENS-D-24-02558R1

Env from EIAV vaccine delicately regulates NLRP3 activation via attenuating NLRP3-NEK7 interaction

PLOS Pathogens

Dear Dr. Wang,

Thank you for submitting your manuscript to PLOS Pathogens. After careful consideration, we feel that it has merit but does not fully meet PLOS Pathogens's publication criteria as it currently stands. Therefore, we invite you to submit a revised version of the manuscript that addresses the points raised during the review process.

Please submit your revised manuscript within 30 days. If you will need more time than this to complete your minor revisions, please reply to this message or contact the journal office at plospathogens@plos.org. Please include the following items when submitting your revised manuscript:

We look forward to receiving your revised manuscript.

Kind regards,

Ulf Dittmer

Academic Editor

PLOS Pathogens

Susan Ross

Section Editor

PLOS Pathogens

Sumita Bhaduri-McIntosh

Editor-in-Chief

PLOS Pathogens

orcid.org/0000-0003-2946-9497

Michael Malim

Editor-in-Chief

PLOS Pathogens

orcid.org/0000-0002-7699-2064

**Additional Editor Comments :**

The authors performed additional experiments and addressed all critical point adequately. However, there are still some minor revision needed:

The experiments in which the authors tried to show the colocalization of NEK7 and NLRP3 by flow cytometry should be removed from text and supplement, as it is technically not valid.

There is still no information on the number of data points that are shown in the bar graphs.

Please add brightfield images to all figures showing flourescence microscopy data.

There is still no information about the number of cells that were infected with 2x10^5 TCID50.

**Journal Requirements:**

1) We have noticed that you have cited Table 1 in the manuscript file but there is no corresponding table in the manuscript.  Please amend your manuscript to include this table noting that tables should not be uploaded as individual files.

2) Please ensure that the funders and grant numbers match between the Financial Disclosure field and the Funding Information tab in your submission form. Note that the funders must be provided in the same order in both places as well. Currently, "the Tianchi Talent Introduction Plan (IWA2023)" is missing from the Funding Information tab.

3) Please add in the legends of Figures (3A,7A, and 9) that they are created with BioRender.com.

**Reviewers' Comments:**

Reviewer's Responses to Questions

**Part I - Summary**

Reviewer #1: The authors have addressed most of my concerns with the previous version, but a few points remain that should be addressed before acceptance.

**Part II – Major Issues: Key Experiments Required for Acceptance**

Reviewer #1: The authors have performed a convincing new experiment using bimolecular fluorescence complementation to prove the colocalization of NEK7 and NLRP3. The previous experiment where they tried to prove the colocalization by flow cytometry should be removed altogether from text and supplement, as it is technically not valid. The cited literature shows a similar dot plot, but I believe they are showing different data in these plots (confocal microscopy data). It is not technically plausible to use flow cytometry - unless it is imaging flow cytometry - to study colocalization.

Again, I highly appreciate the bimolecular fluorescence complementation study, which is convincing and should be the only result shown in the revised manuscript.

**Part III – Minor Issues: Editorial and Data Presentation Modifications**

Reviewer #1: The authors have added information about the number of experiments to this revised manuscript, but there is still no information about the number of data points that are shown in the bar graphs.

Please add brightfield images to all figures showing flourescence microscopy data, at the moment they have only been added to Figure 1E and Figure 5E.

In the Methods section "Infection", there is still no information about the number of cells that were infected with 2x10^5 TCID50.

PLOS authors have the option to publish the peer review history of their article (what does this mean? ). If published, this will include your full peer review and any attached files.

**Do you want your identity to be public for this peer review?** For information about this choice, including consent withdrawal, please see our Privacy Policy .

Reviewer #1: No

**Figure resubmission:**
---

## [Editor Report · Decision Letter 2]

Dear Dr Wang,

We are pleased to inform you that your manuscript 'Env from EIAV vaccine delicately regulates NLRP3 activation via attenuating NLRP3-NEK7 interaction' has been provisionally accepted for publication in PLOS Pathogens.

Best regards,

Ulf Dittmer

Academic Editor

PLOS Pathogens

Susan Ross

Section Editor

PLOS Pathogens

Sumita Bhaduri-McIntosh

Editor-in-Chief

PLOS Pathogens

orcid.org/0000-0003-2946-9497

Michael Malim

Editor-in-Chief

PLOS Pathogens

orcid.org/0000-0002-7699-2064

All remaining points have been convincingly addressed.
---

## [Editor Report · Acceptance letter]

Dear Dr Wang,

We are delighted to inform you that your manuscript, "Env from EIAV vaccine delicately regulates NLRP3 activation via attenuating NLRP3-NEK7 interaction," has been formally accepted for publication in PLOS Pathogens.

Best regards,

Sumita Bhaduri-McIntosh

Editor-in-Chief

PLOS Pathogens

orcid.org/0000-0003-2946-9497

Michael Malim

Editor-in-Chief

PLOS Pathogens

orcid.org/0000-0002-7699-2064